# Porous Structure of β-Cyclodextrin for CO_2_ Capture: Structural Remodeling by Thermal Activation

**DOI:** 10.3390/molecules27217375

**Published:** 2022-10-30

**Authors:** Tianxiang Guo, Runan Zhang, Xilai Wang, Lingfeng Kong, Junpeng Xu, Huining Xiao, Alemayehu Hailu Bedane

**Affiliations:** 1Hebei Key Lab of Power Plant Flue Gas Multi-Pollutants Control, Department of Environmental Science and Engineering, North China Power University, Baoding 071003, China; 2MOE Key Laboratory of Resources and Environmental Systems Optimization, College of Environmental Science and Engineering, North China Electric Power University, Beijing 102206, China; 3Department of Chemical Engineering, University of New Brunswick, Fredericton, NB E3B 5A3, Canada

**Keywords:** CO_2_ adsorption, β-cyclodextrin, thermal activation, structural remodeling, pore formation

## Abstract

With a purpose of extending the application of β-cyclodextrin (β-CD) for gas adsorption, this paper aims to reveal the pore formation mechanism of a promising adsorbent for CO_2_ capture which was derived from the structural remodeling of β-CD by thermal activation. The pore structure and performance of the adsorbent were characterized by means of SEM, BET and CO_2_ adsorption. Then, the thermochemical characteristics during pore formation were systematically investigated by means of TG-DSC, in situ TG-FTIR/FTIR, in situ TG-MS/MS, EDS, XPS and DFT. The results show that the derived adsorbent exhibits an excellent porous structure for CO_2_ capture accompanied by an adsorption capacity of 4.2 mmol/g at 0 °C and 100 kPa. The porous structure is obtained by the structural remodeling such as dehydration polymerization with the prior locations such as hydroxyl bonded to C6 and ring-opening polymerization with the main locations (C4, C1, C5), accompanied by the release of those small molecules such as H_2_O, CO_2_ and C_3_H_4_. A large amount of new fine pores is formed at the third and fourth stage of the four-stage activation process. Particularly, more micropores are created at the fourth stage. This revealed that pore formation mechanism is beneficial to structural design of further thermal-treated graft/functionalization polymer derived from β-CD, potentially applicable for gas adsorption such as CO_2_ capture.

## 1. Introduction

Recently, many researchers have shown an increased interest in global climate change mainly caused by carbon dioxide (CO_2_) emission [1,2,3]. Two kinds of approaches [4] for carbon sequestration are used to reduce the emission of CO_2_. The first one is ecological carbon sequestration [5] based on the plant photosynthesis, and the second alternative is technological carbon sequestration [6] by processing CO_2_ in terms of carbon capture [7], utilization [8,9] and storage [10] (CCUS) technology [11], bioenergy with carbon capture and storage technology (BECCS) [12], and direct air carbon capture and storage technology (DACCS) [13].

Adsorption is one of the most common methods to capture CO_2_ because of its high adsorption capacity, low cost, low energy requirements and ease of operation [14]. According to the literature report, the adsorbents can be divided into three types [15]: low-temperature (<200 degrees Celsius (°C)) adsorbents such as metal-organic frameworks [16] (MOFs), porous organic polymers [17], porous carbons [18], zeolites [19] and organic-inorganic hybrids [20]; intermediate (200–400 °C) absorbents such as metal oxides [21] and hydrotalcite [22], and high-temperature (>600 °C) adsorbents such as lithium zirconate [23]. Porous carbon-based materials have attracted much attention in CO_2_ capture owing to wide availability, physiochemical stability and variable design to tune their porosity [24,25,26,27,28], such as those derived by natural resources including lignin [29,30], starch [31], cellulose [32,33], chitosan [34,35], cyclodextrin [36,37]. Generally, the temperature for physical/chemical activation [27] is found in the range of 400–900 °C. The evaporation of the gas produces a loose structure, limited by the factors of activation temperature and heating rate.

In recent years, cyclodextrins (CDs) have been considered as environmentally friendly adsorptive separation materials due to their green, non-toxic and pollution-free properties in various fields such as food [38], environment [39], medicine [40,41,42], polymer synthesis [43,44,45], cosmetics [46], chemical detection [47] and other fields [48]. It is worth mentioning that the cyclodextrins themself can be used as carriers of CO_2_ gas [49]. However, the inclusion amount of CO_2_ is low. Therefore, those structural remodeling methods using thermal activation [50] and grafting [51,52,53,54] can be used to improve the pore structure of CDs for CO_2_ capture. The cyclodextrin (CD)/graphene composite aerogel synthesized based on hydrothermal reaction at 80 °C for 18 h exhibits an adsorption capacity of CO_2_ at 45 mg/g [55]. γ-cyclodextrin derivative CD-PAF-1 based on dehydration reaction exhibits an adsorption capacity of CO_2_ about 48.6 cm^3^/g [49]. Then, two isostructural cyclodextrin-based CD-MOFs (CD-MOF-1 and CD-MOF-2) are demonstrated to have an inverse ability to selectively capture CO_2_ from C_2_H_2_ by single-component adsorption isotherms and dynamic breakthrough experiments. These two MOFs exhibit excellent adsorption capacity and benchmark selectivity (118.7) for CO_2_/C_2_H_2_ mixture at room temperature [56]. In addition, a new solid acid adsorbent for CO_2_ capture derived from β-cyclodextrin has been obtained and achieves the capacity of 39.87 cm^3^/g at 3.5 bar [57]. The amino functionalization adsorbent NH_2_-β-CD-MOF has also been constructed for the first time, and exhibits great selective adsorption of CO_2_/N_2_ (947.52) [53]. Microporous carbon nanospheres have also been prepared from β-CD by solvothermal carbonization in o-dichlorobenzene with the presence of various concentrations of p-toluene sulfonic acid (PTSA) [58]. For the thermal activation, a rapid temperature-assisted synthesis has been reported to improve the porous structure of the cyclodextrins for CO_2_ adsorption [59]. Regarding the pyrolysis stages [50], apparent activation energy and pyrolysis kinetics of the cyclodextrins [60] have been preliminarily investigated in some literature. Then, the adsorption thermodynamics of CO_2_ on β-CD-derived adsorbent by thermal activation have been reported in our recent paper [61]. However, the pore formation mechanism during the thermal activation of β-CD for CO_2_ uptake has been not systematically investigated so that it is not clear how to further promote the performance of CO_2_ adsorption through improving the preparation method of the adsorbent derived from β-CD.

In this work, the β-CD derivative was prepared based on the structural remodeling of β-CD by thermal activation to obtain a potential adsorbent for CO_2_ capture. The pore formation mechanism during the thermal activation of β-CD was systematically investigated, which is different from the above researches. Then, the activation process and thermochemical characteristics were experimentally analyzed in order to reveal the pore formation mechanism, and the activation pathway was theoretically deduced and the porous structure was characterized. Different techniques were used to characterize the adsorbent including textural analysis: BET, BJH and HK for pore structure; spectroscopy techniques, SEM, EDS, FTIR and XPS for surface morphology; static adsorption method for CO_2_ adsorption performance; in situ TG-FTIR, TG-DSC, TG-MS and DFT for in situ analysis during the thermal activation. Therefore, the general objective of this study is to provide a clear understanding of pore formation mechanism essentially for β-CD derivatives as the adsorbents and provide benefit for process design of thermal-treated graft/functionalization polymer by using β-CD as carbon precursor, which potentially applies for gas adsorption such as CO_2_ capture.

## 2. Results and Discussion

### 2.1. Thermal Activation Process

In order to investigate the thermal activation characteristics of β-CD, the TG curves, DTG curves and DSC curves of β-CD samples under different heating conditions were experimentally obtained under nitrogen atmosphere, and the results are shown in Figure 1. As seen from Figure 1(a-1,a-2), according to the mass loss, the thermal activation process of β-CD can be divided into four stages: drying stage (first stage), stable stage (second stage), main pyrolysis stage (third stage) and restructuring stage (fourth stage). The obvious mass loss appears at the drying stage and main pyrolysis stage. It is worth mentioning that the temperature ranges of the four stages are not fixed depending on the heating rate. When the heating rate increases, the temperature range of every stage moves towards high temperature region. For instance, the four thermal activation stages of β-CD sample with a heating rate of 5 °C/min could be divided by the drying stage below 115 °C, the stable stage between 115 °C and 237.9 °C, the main pyrolysis stage between 237.9 °C and 410 °C, the restructuring stage above 410 °C. However, the end temperature of the drying stage increases from 115 °C to 154 °C when the heating rate increases from 5 °C/min to 20 °C/min.

As shown in Figure 1(a-3,a-4), an obvious endothermic peak appears at the first stage, which is ascribed to the escape process of those small adsorbed molecules such as water [62]. At the second stage, the mass of the sample remains almost unchanged. Then, the sample loses most of its mass at the third stage, accompanied by a tendency of first the endothermic peak then the exothermic peak, which may result from a cooperative effect of those cracking reactions such as ring cleavage and the oxidation reactions such as the formation reactions of the ether, aldehyde and ketone. The fourth stage is mainly related to the structural remodeling such as graphitization based on an endothermic process, accompanied by a small amount of mass loss.

In order to further investigate the thermochemical characteristics at the third stage, the extra TG-DSC graphs at the final activation temperatures of 290 °C, 310 °C and 330 °C were obtained. In the experiments, the β-CD samples were firstly heated from room temperature to 180 °C with a heating rate of 5 °C/min and kept for 15 min with the purpose of avoiding the interferences of the adsorbed small molecules such as water for thermochemical analysis at the third stage. The results are shown in Figure 1(b-1–b-4). As seen from TG curves in Figure 1(b-1), the end temperature of the second stage/the initial temperature of the third stage is found to be about 237.9 °C, and an endothermic peak is verified to exist between 180 °C and 270 °C. Unfortunately, some sharp interference peaks (at 180 °C, 270 °C, 290 °C, 310 °C, 330 °C) possibly appear in the DSC curves when the heating environment changes. In order to distinguish the pyrolysis peak from the interference peak, the extra TG-DSC graphs at the final activation temperatures of 290 °C, 310 °C and 330 °C were obtained, respectively, and the results are shown in Figure 1(b-2–b-4). As shown in Figure 1(b-2–b-4), the mass loss of β-CD sample increases with an increase in the final activation temperature from 290 °C to 330 °C, meaning that the thermochemical reactions happen more violently at a higher activation temperature. Interestingly, comparing Figure 1(b-2–b-4) with Figure 1(b-1), the sharp peak at 270 °C on DSC curve disappears, indicating that these interference peaks (at 180 °C, 270 °C, 290 °C, 310 °C, 330 °C) really exist when the heating environment changes. Comparing Figure 1(b-1) with Figure 1(b-2), there should be at least an extra endothermic peak appearing above 270 °C except the endothermic peak between 180 °C and 270 °C. Comparing Figure 1(b-2) with Figure 1(b-3) and Figure 1(b-3) with Figure 1(b-4), at least two endothermic peaks are verified to appear between 180 °C and 330 °C, and there should be at least two kinds of endothermic reactions such as glycosidic bond cleavage and ring-opening reaction of pyranose happening at the third stage, accompanied by obvious mass loss due to the release of water and carbon oxides such as CO and CO_2_ [63]. 

Meanwhile, the linear regression analysis of solid-state thermal decomposition model function g(α) via activation temperature at the third stage was performed, and the results are shown in Table 1. As shown in Table 1, the thermal activation of β-CD is considered as a diffusion control process, accompanied by the better correlation coefficients for those D solid state thermal decomposition model functions of D_1_, D_2_, D_3_ and D_4_ [64]. Therefore, the Jander function D_3_ based on three-dimensional diffusion control step is considered to be the best reasonable way to describe the thermochemical activation behaviors of β-CD. The obtained average apparent activation energy is 110.5 kJ/mol.

### 2.2. Thermochemical Characteristics

#### 2.2.1. Gas Product

Figure 2 presents the analysis results of in situ TG-FTIR and TG-MS with the purpose of detecting the gas composition during β-CD activation. As seen from in situ TG-FTIR presented in Figure 2(a-1,b-1), the gases of H_2_O, CO and CO_2_ are detected during β-CD activation with a final activation temperature of 800 °C regardless of heating rate. The existence of the knee of the release curve of water at a heating rate of 5 °C/min indicates that there are at least two kinds of water release mechanisms during β-CD activation, possibly for the adsorbed water and the structural water. Moreover, the structural water should be released at both the third stage and the fourth stage, accompanied by an amount increasing tendency at a given heating rate with an increase in activation temperature but an amount decreasing tendency at the same activation temperature with the increase in heating rate. Therefore, the high activation temperature and low heating rate are advantageous for the dehydration reactions during the adsorbent preparation. The dehydration advantage of lowering the heating rate is ascribed to the smaller temperature gradient between the inner and the surface of the sample. In addition, the release amount of CO increases with an increase in activation temperature, but almost the same tendency remains regardless of whatever the heating rate is. This indicates that the release of CO is mainly affected by the activation temperature. In contrast, the release amount of CO_2_ increases with the increase in both activation temperature and heating rate, indicating that the high activation temperature and high heating rate are advantageous for the release of CO_2_ during β-CD activation. The phenomenon is also illustrated by the infrared spectral peak near 2300 cm^−1^, accompanied by the peak intensity increase with an increase in activation temperature but the decrease of heating rate.

In addition, given the fact that there is no obvious absorption band between 1000 cm^−1^ and 1300 cm^−1^, the quantities of those gas organic products containing the group of C-O and C-C in chain molecule are few, so the possible gas organic products should exist in the form of unsaturated hydrocarbon or ring compound, which is also verified based on the peaks near 1539 cm^−1^ and 1695 cm^−1^ (the ring skeletal vibration and the stretching vibration of unsaturated bonds such as C=C and C=O). Meanwhile, the very weak peak at the high wavenumber side of 3000 cm^−1^ indicates that the release amounts of those possible unsaturated hydrocarbons or ring compounds should be slight. The weakening tendency of the peak intensity near 1539 cm^−1^ and 1695 cm^−1^ shows that the release amounts of possible unsaturated hydrocarbons or ring compounds decrease with an increase in the heating rate, which is possibly related to the release amount increase in CO_2_ with the increase in the heating rate during β-CD activation.

In order to detect those small organic molecules, Figure 2(a-2,b-2) exhibits the group fragments of gas products based on TG-MS. As can be seen, the appeared MS peaks at *m*/*z* of 18, 28 and 44 are ascribed to the existence of H_2_O, CO and CO_2_ in the gas products, respectively. Then, the MS peaks at *m*/*z* of 1 and 17 are related to the groups of ^1^H and ^16^OH mainly derived from the decomposition of H_2_O under the effect of ion source from MS equipment. There is no MS peak observed at *m*/*z* of 12, indicating that the molecule of CO does not decompose during the MS observation process. In consideration of the existence of the MS peak at *m*/*z* of 16 related to ^16^O, the molecule of CO_2_ possibly decomposes during the MS observation process so that the MS peak intensity at *m*/*z* of 44 is weaker than that at *m*/*z* of 28, although the release amount of CO_2_ in the gas products is generally larger than that of CO under the same thermal conditions (see Figure 2(a-1,b-1)). These secondary ions of ^1^H, ^16^OH and ^16^O contribute to the appearance of the MS peaks at *m*/*z* of 29, 30, 32 and 34 related to those possible groups of ^12^CH^16^O, ^12^CH_2_^16^O, ^16^O_2_ and H_2_^16^O_2_. Subsequently, the existence of the strong MS peak at *m*/*z* of 40 and the weak MS peak at *m*/*z* of 80 indicates there should be at least one kind of small organic molecule with a possible molecular mass of 40 or 80 appearing in the gas products. The possible molecules are cyclopropene (^12^C_3_H_4_), cyclohexadiene (^12^C_6_H_8_) or cyclopentadienone(^12^C_5_H_4_^16^O). The cracking of the cyclopropene contributes to the appearance of the MS peaks at *m*/*z* of 14 (–^12^CH_2_–) and 38 (–^12^C_3_H_2_–). Cyclopropene can be also considered as the direct cracking product of cyclohexadiene. Then, cyclopentadienone can crack and form the active and unstable groups of –^12^C_3_H_3_ and –^12^CH=^12^C=O, which further form stable groups of ^12^C_3_H_4_ and ^12^CH_3_C^12^HO combining with proton transfer, accompanied by the contribution to the MS peaks at *m*/*z* of 40 and 44. In addition, the existence of relatively strong peaks at *m*/*z* of 20 and 36 means that the isotopic atoms of ^12^C, ^1^H and ^16^O probably appear in those gas molecules and ions, especially for the isotopic atom of ^18^O. According to the stability and abundance ratio of isotopes, hydrogen atoms (^1^H–99.989%, ^2^D–0.011%), carbon atoms (^12^C–98.93%, ^13^C–1.07%), oxygen atoms (^16^O–99.76%, ^18^O–0.21%) are considered for the MS analysis. Moreover, only one isotopic atom is considered to exist in given molecule or ion. Then, their contributions for those MS peaks are listed in Table 2.

#### 2.2.2. Solid Product

In order to explore the changes of chemical components of β-CD samples after thermal activation, the surface functional groups of solid samples after thermal activation were characterized by means of EDS, FTIR and XPS, then the results are shown in Figure 3. As shown in Figure 3a, the content proportion of the elemental carbon and elemental oxygen on the surface of original β-CD sample is 78:22, which is higher than that theoretically in the bulk phase (C_42_H_70_O_35_, 9:10) of the original β-CD sample. This means that the oxygen atom mainly tends to locate towards the inner of the β-CD aggregates while the carbon atom tends to locate towards the outer of the β-CD aggregates due to the existence of the hydrogen bonding. Then, the content proportion of the element C to element O on the surface increases to 87:12, 94:7 and 98:2 when the final activation temperature is up to 400 °C, 600 °C and 800 °C, respectively. This indicates that those small oxygen-containing molecules such as water (H_2_O) and carbon oxides (CO_2_ or CO) are released during β-CD activation and the total release amount of those small oxygen-containing molecules increases with an increase in activation temperature, just like those shown in Figure 1(a-1,b-1).

As seen from Figure 3b, the surface functional groups are similar for those β-CD samples activated at different final activation temperatures. The strong band peak near 3400 cm^−1^ is ascribed to the stretching vibration of O-H groups from adsorbed water or residual hydroxyls after activation. The band with two peaks of 1631 cm^−1^ and 1604 cm^−1^ is mainly assigned to the in-plane bending vibration of O-H group and stretching vibration of C=C bond. The band with two peaks of 1385 cm^−1^ and 1353 cm^−1^ may be caused by the ring skeletal vibration and in-plane bending vibration of C-H group. The band between 1000 cm^−1^ and 1300 cm^−1^ with two peaks 1125 cm^−1^ and 1098 cm^−1^ possibly results from the antisymmetric stretching vibration of the C-C-C/C-O-C groups and the stretching vibration of the C-O-H groups on the linear chain. Then, secluded CO and CO_2_ absorbed result in the appearance of the peak bands between 2000 cm^−1^ and 2100 cm^−1^, between 2300 cm^−1^ and 2400 cm^−1^, respectively.

As shown in Figure 3c,d, the X-ray photoelectron spectra of carbon element of the β-CD samples after activation above 400 °C possibly consist of eight child peaks. The right three child peaks are related to C1s bonded to carbon atom (C=C: 284.3 eV, graphite C: 284.8 eV and C-C: 285.3 eV). The left five child peaks are related to C1s bonded to oxygen atom (O-C*-O: 285.8 eV; O-C*-C: 286.4 eV; C*=O from aldehydes-287 eV, ketones-287.7 eV and carboxyls-289.5 eV). The carbon element on the sample after activation tends to finally exist in the form of graphite C according to the increase in phenomenon of graphite C content with an increase in the activation temperature from 600 °C to 800 °C, accompanied by the content decrease in C-C and C=C. In addition, the intensity increase in both the peaks at 287.7 eV and 289.5 eV represents the increase in surface groups of carboxyls and carbonyls when the activation temperature rises from 600 °C to 800 °C, and the phenomenon can be also verified by the intensity increase in the peak at 532.7 eV related to C=O* from carboxyls and carbonyls based on the X-ray photoelectron spectra of oxygen element shown in Figure 3e,f. These surface group sites C=O* from carboxyls and carbonyls possibly contribute to the release of carbon oxides such as CO and CO_2_. As is also seen from Figure 3e,f, the intensity decrease in the peak at 533.6 eV related to C-OH means the existence of intramolecular or intermolecular dehydration reactions, which happen more violently with an increase in the activation temperature. Then, the weakening intensity of the peak at 533.8 eV related to C-O-C and O*-C=O indicates the cleavage of ether bond at the locations of C1, C4 or C5, which possibly contributes to the formation of some carboxyls and carbonyls when the activation temperature rises from 600 °C to 800 °C. Meanwhile, some carboxyls and carbonyls are possibly derived from those aldeyhydes accompanied by the intensity decrease in the peak at 531.5 eV related to aldeyhydes; some ring compounds are formed. Then, the intensity increase in the peak at 534.5 eV possibly related to C-O* bonded to aromatic rings should indicate the formation of the pore wall on the sample after activation.

### 2.3. Structural Remodeling

#### 2.3.1. Fracture Process of Glycosidic Bond

So far, there are three main views on the formation path about glycosidic bond cleavage: homolysis (free radical reaction), heterolysis, and synergistic mechanism [65]. With the aim to demonstrate the path of glycosidic bond cleavage, DFT calculation at the same level was performed in this section. The convergence criteria for energy, force and displacement are 2 × 10^−5^ Hartree (Ha), 0.004 Ha^−1^ and 0.005 Å, respectively. The molecular structures of cellobiose and β-CD used for calculation and analysis are shown in Figure 4a,b. There are two possibilities for the homogeneous and heterogeneous cleavage positions of β-1,4-linked glycosidic bond, namely C1’-O/C4-O bonds. Therefore, five reaction paths have been calculated and analyzed in this section. The activation energy of homolysis, heterolysis, and the synergistic reaction of β-CD glycosidic bond were deduced, and the results are presented in Figure 4c.


Homogeneous cleavage of C1’-O bond: since there is no saddle point on the potential energy surface, it is difficult to determine the transition state of the homolytic reaction, and the bond dissociation energy (BDE) is generally used as the activation energy for such reactions. The BDE equation of cracking reaction is listed as the following equation:
(1)BDE(A−B)=H(A· )+H(B· )−H(A−B)
where *H*(*A* − *B*), *H*(*A*·) and *H*(*B*·) are the enthalpy values of the molecules *A* − *B* and the free radicals *A* and *B* generated by its fracture, respectively. The calculated energy barrier of this path is 343.9 kJ/mol.C4-O bond cleavage: the energy barrier is 341.3 kJ/mol.Isocracking of C4-O bond: due to the electronegativity difference between carbon atom and oxygen atom, the possible position of glucosidic bond is the same as that of homolytic reaction. The calculation method of activation energy is the same as that of homolytic reaction, and the energy barrier of this reaction path is 357.5 kJ/mol.Heterosis of C1’-O bond: energy barrier is 498.8 kJ/mol.Synergistic reaction mechanism: in this path, the glycosidic bond exhibits a six-membered ring transition state, and the energy barrier of this path is 257.3 kJ/mol.


According to the above calculation, the heterolysis of glycosidic bond requires higher energy compared with the homolysis of glycosidic bond; then, the synergistic reaction mechanism exhibits the lowest energy barrier, and most probably occurs during the activation of β–CD. In contrast, the dissociation energy of C1’–O bond is higher than that of C4–O bond, indicating that the glycosidic bond cleavage happens primarily at the location of C4–O.

#### 2.3.2. Dehydration of Pyranose

In consideration of the fact that β-CD is mainly composed of pyranose, the intramolecular dehydration of β-CD is simplified as the intramolecular dehydration of pyranose. Here, the intramolecular dehydration related to the cleavage of glycosidic bonds is not considered. According to the hydroxyl structure of pyranose, seven dehydration paths were deduced, and the results are presented in Figure 4d. The constructed structures were structurally optimized based on Materials Studio, and the transition states were searched by the B3LYP function based on density functional theory [66,67] (DFT) in the DMOL3 module; then, the activation energy of each reaction path was obtained. The obtained activation energies are 344.9 kJ/mol for C1(OH) and C2(H) dehydration, 397.8 kJ/mol for C2(OH) and C1(H) dehydration, 378.3 kJ/mol for C2(OH) and C3(H) dehydration, 443.9 kJ/mol for C3(OH) and C2(H) dehydration, 351.3 kJ/mol for C3(OH) and C4(H) dehydration, 354.4 kJ/mol for C4(OH) and C3(H) dehydration, and 76.3 kJ/mol for C6(OH) and C5(H) dehydration, respectively. The real activation energy of pyranose dehydration should be between 76.3 kJ/mol and 443.9 kJ/mol. The dehydration reaction between C6(OH) and C5(H) exhibits the lowest activation energy, demonstrating that the dehydration reaction of pyranose can easily occur in the position of C6 hydroxyl group. The dehydration reaction between C3(OH) and C2(H) exhibits the highest activation energy due to the fact that the groups of C3(OH) and C2(H) are distributed on two sides of the C3-C2 bond on the pyran ring. In the pyranose ring, compared with the other paths except the pathway 7, the dehydration pathway of C1(OH) and C2(H) exhibits relatively lower activation energy, and the hydroxyl group on C1 is relatively easy to detach for generating water because of the repulsive force between the ether bond on pyran ring. Therefore, the dehydration at the locations of C6(OH) and C1(OH) can act as a bridging among the different pyran rings, and plays a role of the skeleton of the porous structure.

Nevertheless, the cleavage of glycosidic bond also possibly occurs at the location of C1, just like that shown in Figure 4c. Combined with the analysis of the current literature [68], the product of glycosidic bond cleavage by this way might be as high as 50% (levoglucosan) during the activation process. As a result, the ring-opening reaction of β-CD at the location of C1 happens as the competing process of the C1 dehydration reaction.

#### 2.3.3. Opening Ring of Glucose Unit

In order to reveal the possible ring-opening reactions, the mass spectrum of original β-CD sample was obtained, and the results are shown in Figure 4e. Then, the possible ring-opening reactions were analyzed, and the results are illustrated in Figure 4f. As shown in Figure 4e, the peak near 1169 represents the isotope molecular ion peak of β-CD (relative molecular mass, 1134), accompanied by possible existed groups such as hydroxyls (relative mass, 17) derived from the dissociation of included water by β-CD during ionization process. Comparing the main MS peaks at *m*/*z* of 183, 447, 559, 600, 713 and 826 with theoretical MS peaks at *m*/*z* related to those complete glucopyranose units (see Table 3), the obvious differences among them indicate that there is more than one path for the ring-opening reactions except the glycosidic bond cleavage at the location of C4-O. The appearance of the MS peak at *m*/*z* of 826 means the existence of the glycosidic bond cleavage at the location of C1-O based on the homolysis of glycosidic bond (see Figure 4f). In addition, the MS peak at *m*/*z* of 183 possibly results from the combination of the homolysis of glycosidic bond and ring-opening reaction at the ether bond of the pyran ring, accompanied by proton transfer (e.g., one glucopyranose unit containing one ^13^C atom, accompanied by one ^16^O atom at the location of C4 derived from the homolysis of glycosidic bond and the transfer of four protons). The possible residual fragment M_146_ derived from one glucopyranose unit by losing the ^16^O atom at the location of C1 can further crack into the fragment M_73_ based on two kinds of the ring-opening reactions at the locations of C1 and C4. Then, the fragment M_73_ is bonded to the fragment M_486_ (three complete glucopyranose units based on the synergistic reaction mechanism shown in Figure 4c) and forms the strongest MS peak at *m*/*z* of 559. The MS peaks at *m*/*z* of 713 and 600 result from the cracking of M_826_, accompanied by the removal of one or two group fragment M_113_ (C_5_H_5_O_3_). The latter can be considered to consist of the group fragments M_73_ and M_40_. Then, the MS peak at *m*/*z* of 447 results from the cracking of M_486_, accompanied by the removal of one group fragment M_39_ (–C_3_H_3_). Then, the other MS peaks at *m*/*z* below 559 are ascribed to the bonding of M_39_ (–C_3_H_3_) and M_40_ (C_3_H_4_) to different glucopyranose units such as the group fragment M_202_. 

#### 2.3.4. Activation Pathway and Porous Structure Formation

In order to reveal the pore formation mechanism during β-CD activation, the activation pathway was inferred based on the above analysis and the related literature [69], and the results are shown in Figure 5. As shown in Figure 5, the activation step at the first stage is attributed to the removal of adsorbed small molecules such as water, followed by a thermally stable state at the second stage. By this time, the pore structure consists of the surface macroporous structure of β-CD aggregates and the microporous structure of the cage cavity in the inner of β-CD molecule. According to the analysis of TG-DSC shown in Figure 1, there are at least two kinds of reactions during subsequent activation at the third stage and fourth stage. The first one is considered to be related to the dehydration reactions including intermolecular dehydration and intramolecular dehydration. The intermolecular dehydration results in the skeleton formation of porous structure by means of the dehydration polymerization (e.g., the dehydration between those hydroxyls bonded to C6), while the intramolecular dehydration tends to result in the graphitization or aromatization of the sample and forms the pore wall of porous structure by means of cyclodehydration. The other one is related to the glycosidic bond cleavage and pyranose ring-opening polymerization [70], accompanied by the removal of carbon oxides. These dehydration reactions, glycosidic bond cleavage reactions and ring-opening reactions result in the structural remodeling of β-CD aggregates, and form the rich microporous structure which is advantageous for CO_2_ capture.

### 2.4. Porous Structure

#### 2.4.1. Surface Morphology

In order to investigate the surface morphology change of β-CD derivatives (β-CD samples after activation), the SEM images of β-CD and its derivatives obtained at the final activation temperatures of 400 °C, 600 °C and 800 °C were produced with a heating rate of 5 °C/min, respectively (Figure 6). As seen from the SEM images at the scales of 1μm and 100 nm in Figure 6a–h, compared with the original β-CD (Figure 6a,b), the β-CD derivative (Figure 6c–h) exhibits a regular porous structure. Moreover, with increase in the final activation temperature from 400 °C to 800 °C, the richer porous structure seems to be formed.

#### 2.4.2. Pore Size Distribution

In order to verify the formation of the richer porous structure at relatively higher final activation temperature, the pore structural characteristics of the β-CD derivatives were activated at 600 °C and 800 °C with a heating rate of 5 °C/min. The data were obtained based on methods such as BET, and the results are shown in Table 4.

It can be seen from Table 4 that both the specific surface area and pore volume of the β-CD derivatives are obviously improved compared with the original β-CD sample before activation. Taking the sample at a final activation temperature of 500 °C as an example, its specific surface area and pore volume increase about 5 times and 2.5 times compared with the original β-CD, accompanied by 7 times increase in micropore volume. Moreover, the β-CD derivatives exhibit the rich microporous structure, which occupied 70% of total pore volume at the final activation temperature of 500 °C and up to 90% of total pore volume at a final activation temperature of 800 °C, which is advantageous for CO_2_ capture. Interestingly, with increasing the final activation temperature from 500 °C to 800 °C, both the specific surface area and the total pore volume increase almost 4 times but the average pore size only decreases from 2.46 nm to 1.78 nm, accompanied by the pore size transfer from mesopore to micropore. More interestingly, both most probable pore sizes of these two β-CD derivatives are almost the same, which is near 0.6 nm. This indicates that the thermal activation improves the porous structure of the β-CD samples, and the increase in final activation temperature from 500 °C to 800 °C results in the formation of more micropores due to the better structural remodeling. 

#### 2.4.3. Performance of CO_2_ Adsorption

Figure 7 shows the adsorption isotherms of CO_2_ on β-CD derivatives obtained by the activation of β-CD at different final activation temperatures and heating rates. As seen from Figure 7a, adsorption capacity of CO_2_ on the derivative from β-CD activation at 500 °C is obviously increased compared with the original β-CD sample (<0.05 mmol/g at 100 kPa with the adsorption temperature less than 25 °C). This indicates that thermal activation of β-CD is beneficial for promoting the capacity of CO_2_ adsorption. The capacity on the β-CD derivative increases with an increase in adsorption pressure but decreases with increasing the adsorption temperature, just like the other carbon-based porous materials we prepared before [71]. This rule is also verified by adsorption capacity change of CO_2_ on the derivative from β-CD activation at 800 °C (see Figure 7b). 

As shown in Figure 7b, the derivative from β-CD activation at 800 °C exhibits an adsorption capacity of 4.2 mmol/g at 0 °C and 100 KPa, which is increased by 91% compared with that on the derivative activated at 500 °C (2.2 mmol/g). This means that the increase in final activation temperature seems to promote the capacity of CO_2_ adsorption on the β-CD derivative; Figure 7c presents the isotherms of CO_2_ adsorption on those β-CD derivatives from different final activation temperatures for 3 h when the adsorption temperature is 0 °C. Figure 7c presents the capacities of CO_2_ adsorption on β-CD derivatives at 100 kPa are 1.7 mmol/g, 2.2 mmol/g, 2.6 mmol/g, 3.4 mmol/g and 4.2 mmol/g, respectively, when the final activation temperatures are 400 °C, 500 °C, 600 °C, 700 °C and 800 °C. Therefore, the fact that the capacity of CO_2_ adsorption on the β-CD derivative at given adsorption temperature and pressure increases with increasing the final activation temperature from 400 °C to 800 °C is verified. The fact also reveals that the adsorption performance of CO_2_ may be relative to the heating capacity of the β-CD sample. The isotherms of CO_2_ adsorption on those β-CD derivatives with a given final activation temperature but different heating rates are presented in Figure 7d. As presented in Figure 7d, adsorption capacity of CO_2_ on the derivative from β-CD activation at a given final temperature such as 500 °C or 700 °C is decreased by about 16.7% at 100 kPa with increasing the heating rate from 5 °C/min to 10 °C/min. These results indicate that decreasing the heating rate is advantageous for promoting the adsorption performance at a given final activation temperature (which allows for more heat supply for the β-CD sample), and so is increasing the final activation temperature at a given activation time. Therefore, it is concluded that increasing the heating capacity will improve the adsorption performance of CO_2_ on the β-CD sample, and the details have been discussed in our previous paper [61].

In addition, as also seen from Figure 7a–d, CO_2_ adsorption on the β-CD derivative is a predominantly physical adsorption process in consideration of the consistency between adsorption isotherm and desorption isotherm regardless of adsorption temperature, final activation temperature and heating rate. 

## 3. Materials and Methods

### 3.1. Thermal Activation

β-CD purchased from Beijing Shuangxuan Microbial Medium Co., Ltd. (Beijing, China) was used to experimentally prepare the β-CD derivate as an adsorbent of CO_2_ based on the thermal activation in a high temperature tube furnace under nitrogen atmosphere, accompanied by the final activation temperatures below 800 °C with different heating rates. In the experiments, the mass of β-CD sample was set as 20 ± 2% mg. The experiments were first carried out in the temperature range from 30 °C to 800 °C with the heating rates of 5 °C/min, 10 °C/min, 15 °C/min and 20 °C/min, respectively. In order to further reveal the thermochemical characteristics, the extra thermal activation experiments were performed, in which the β-CD samples were first heated from room temperature to 180 °C with a heating rate of 5 °C/min and kept for 15 min, then heated to 290 °C, 310 °C and 330 °C with a heating rate of 20 °C/min and kept for 3 h. Subsequently, the thermal activation process was characterized and analyzed by those methods such as in situ thermogravimetric infrared (in situ TG-FTIR, STA 449 F5/TENSOR II, Netzsch/Bruker, Selb, Germany), thermogravimetry-differential scanning calorimetry (TG-DSC, STA 449 F3, Netzsch, Selb, Germany), thermogravimetric mass spectrometry (TG-MS, Pyris Diamond/OmniStar, PerkinElmer/Pfeiffer Vacuum, Waltham, America) and the density functional theory analysis (DFT). The pore formation mechanism of β-CD during thermal activation was discussed based on various characteristics and the model of cellobiose to reduce running time during the DFT calculation. 

### 3.2. Characteristics

In order to reveal the porous structure of the obtained adsorbent, its surface morphology and groups were characterized by means of scanning electron microscope (SEM, QUANTA F250, FEI company field emission, Hillsboro, America), Fourier transform infrared spectroscopy (FTIR, NICOLET 560, Thermo Electron Corporation, Waltham, America), Energy Dispersive Spectroscopy (EDS, Horiba 7021-H), and X-ray photoelectron spectroscopy (XPS, Axis Supra, Shimadzu, Kyoto, Japan). Then, the porous structure including specific surface area, pore volume and pore size distribution were characterized by means of methods such as BET, BJH and HK based on static nitrogen adsorption and desorption (N_2_@ −196.15 °C) on static adsorption instrument (JW-BK122W, Beijing Jingwei Gaobo Technology Co., Ltd., Beijing, China). Subsequently, the adsorption performance of CO_2_ was characterized based on the adsorption isotherms at 0 °C, 25 °C and 50 °C based on the β-CD derivates prepared under different thermal activation conditions. Prior to the adsorption measurements, the β-CD derivates were preliminarily outgassed in vacuum for 8 h at 200 °C.

### 3.3. DFT Calculation

With a purpose of theoretical analysis for the thermal activation process, the model compound cellobiose was set as the basic ring unit of β-CD to reduce the running time during the DFT calculation. Here, the required structure was built and optimized by the ChemDraw software and the software of Materials Studio, respectively. The transition states were analyzed by density functional theory (DFT) based on B3LYP functions in the DMOL3 module [72]. Then, the cases were optimized at B3LYP/6-31G (d, p) level [73] with a series of different basis sets, including the geometries of reactants, intermediates, transition states, and products. The convergence criteria for energy, force, and displacement were 2 × 10^−5^ Hartree (Ha), 0.004 Ha^−1^ and 0.005 Å, respectively. In addition, the SCF tolerance was 1 × 10^−5^ Ha. The transition states were determined by the TS method, and the intrinsic reaction coordinate (IRC) analysis was carried out towards two directions for ensuring that the transition states were related to the correct reactants and products. All optimized structures were determined by frequency analysis, and the virtual frequency of the transition state was selected as the exact one. Then, the frequency analysis and the IRC analysis were performed under the same optimization basis. Finally, the activation energy (reaction barrier) was obtained to analyze the possible path of thermal activation and further reveal the pore formation mechanism.

## 4. Conclusions and Future Perspectives

The investigation indicates that an excellent adsorbent for CO_2_ uptake with adsorption capacity of 4.2 mmol/g at 0 °C and 100 kPa can be obtained via the thermal activation of β-CD. The corresponding process can be divided into four stages; the rich porous structure is mainly formed during the third and fourth stage, accompanied by the spatial structural remodeling and the release of some small molecules such as H_2_O and CO_2_. During the spatial structural reorganization, the bond cracking is considered to mainly happen at the location positions of C4, C1, and C5, respectively. The glycosidic bond cleavage happens in the forms of both the homolysis and synergistic mechanisms. Reactions related to the dehydration polymerization and ring-opening polymerization result in the skeleton formation of porous structure, while residual pyran rings and new rings derived from the cyclodehydration form the pore walls of the porous structure. Then, the better spatial structural remodeling can be obtained by increasing the final activation temperature and decreasing the heating rate, accompanied by the higher capacity of CO_2_ adsorption at the given adsorption conditions. The results are beneficial for processing the design of the thermal-treated graft/functionalization polymer by using β-CD as carbon precursor, which potentially applies to gas adsorption such as CO_2_ capture.

## Figures and Tables

**Figure 1 molecules-27-07375-f001:**
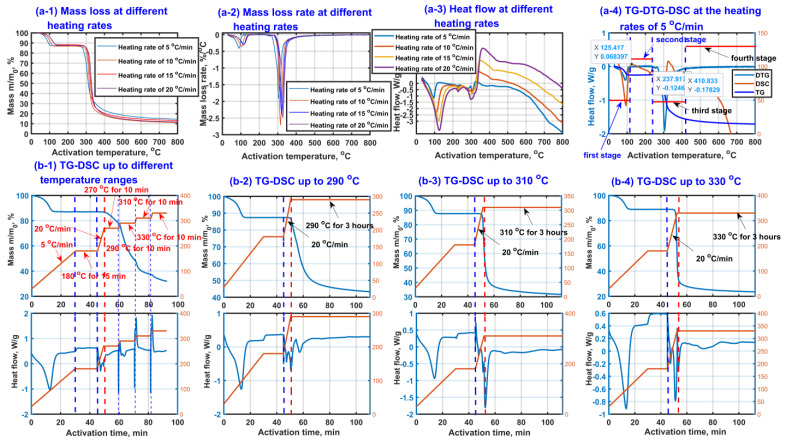
Thermal activation of β-CD samples at different heating conditions (TG curve diagram (**a-1**), DTG curves (**a-2**), DSC curves (**a-3**), TG-DTG-DSC at heating rate of 5 °C/min (**a-4**)) and TG-DSC graphs (heated to 180 °C with a heating rate of 5 °C/min and kept for 15 min, then heated to different temperatures (**b-1**), 290 °C (**b-2**), 310 °C (**b-3**) and 330 °C (**b-4**) with the heating rate of 20 °C/min and kept for 3 h).

**Figure 2 molecules-27-07375-f002:**
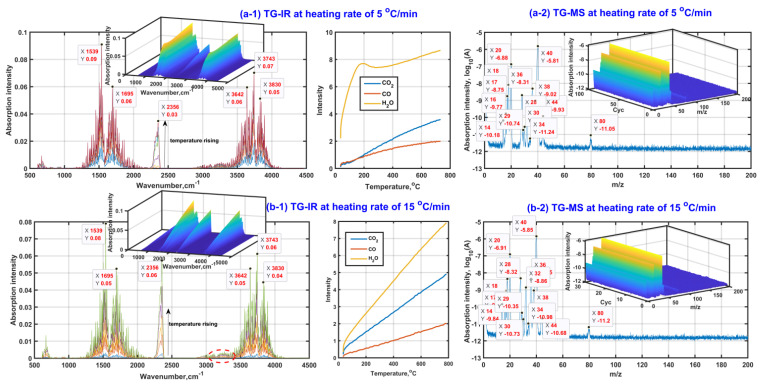
Gas product analysis ((**a-1**,**b-1**) based on in situ TG-FTIR; (**a-2**,**b-2**) based on in situTG-MS).

**Figure 3 molecules-27-07375-f003:**
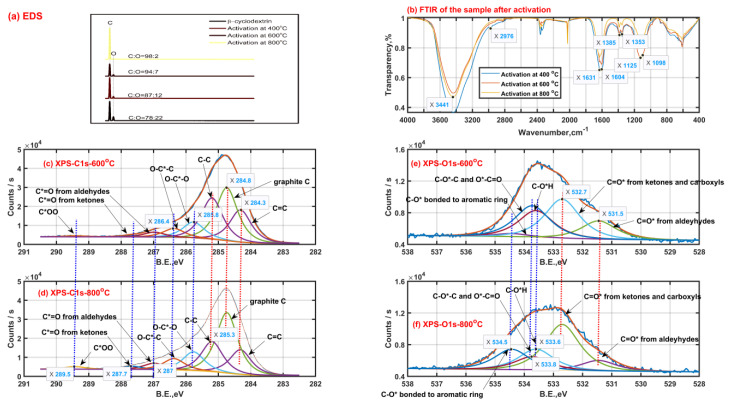
Solid product characteristics with a heating rate of 5 °C/min (EDS Energy dispersive spectroscopy (**a**); Fourier transform infrared spectra (**b**); X-ray photoelectron spectra of carbon ((**c**) −600 °C, (**d**) −800 °C) and oxygen ((**e**) −600 °C, (**f**) −800 °C)).

**Figure 4 molecules-27-07375-f004:**
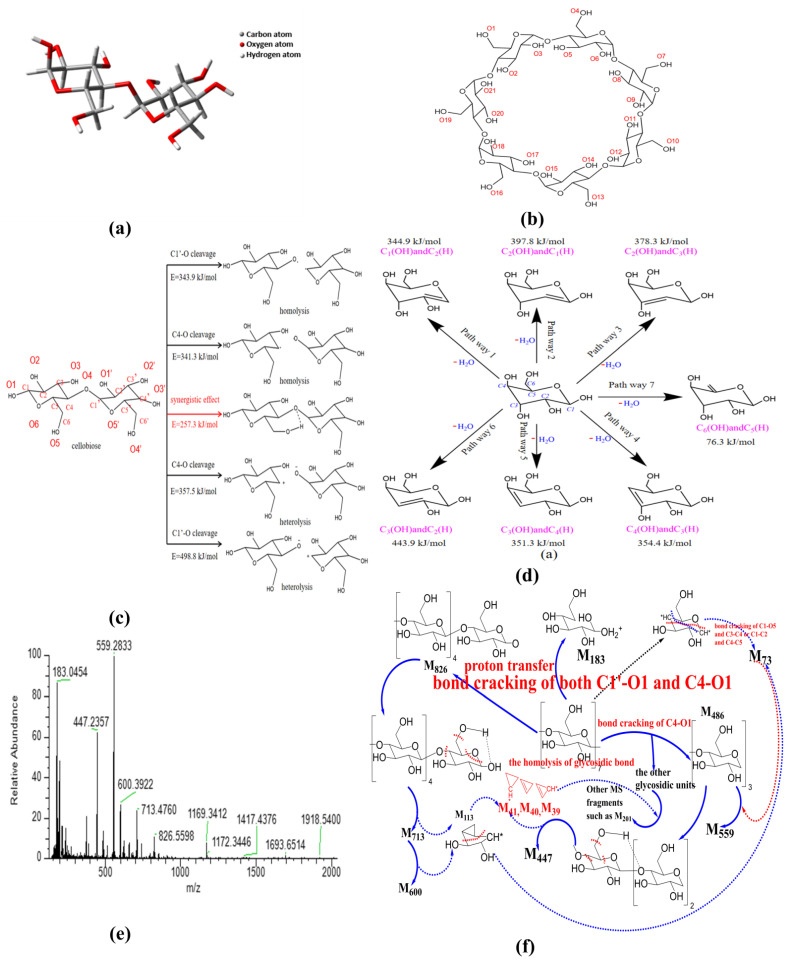
Mechanism analysis of thermal activation of β-CD ((**a**) cellobiose structure; (**b**) β-CD structure; (**c**) fracture process of glycosidic bond; (**d**) dehydration of pyranose; (**e**) MS spectra; (**f**) cracking of β-CD).

**Figure 5 molecules-27-07375-f005:**
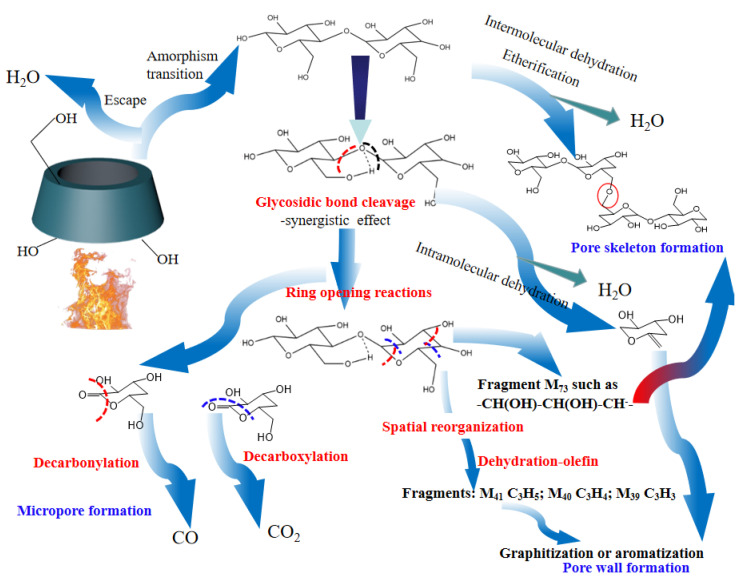
Pore formation mechanism of β-CD during thermal activation.

**Figure 6 molecules-27-07375-f006:**
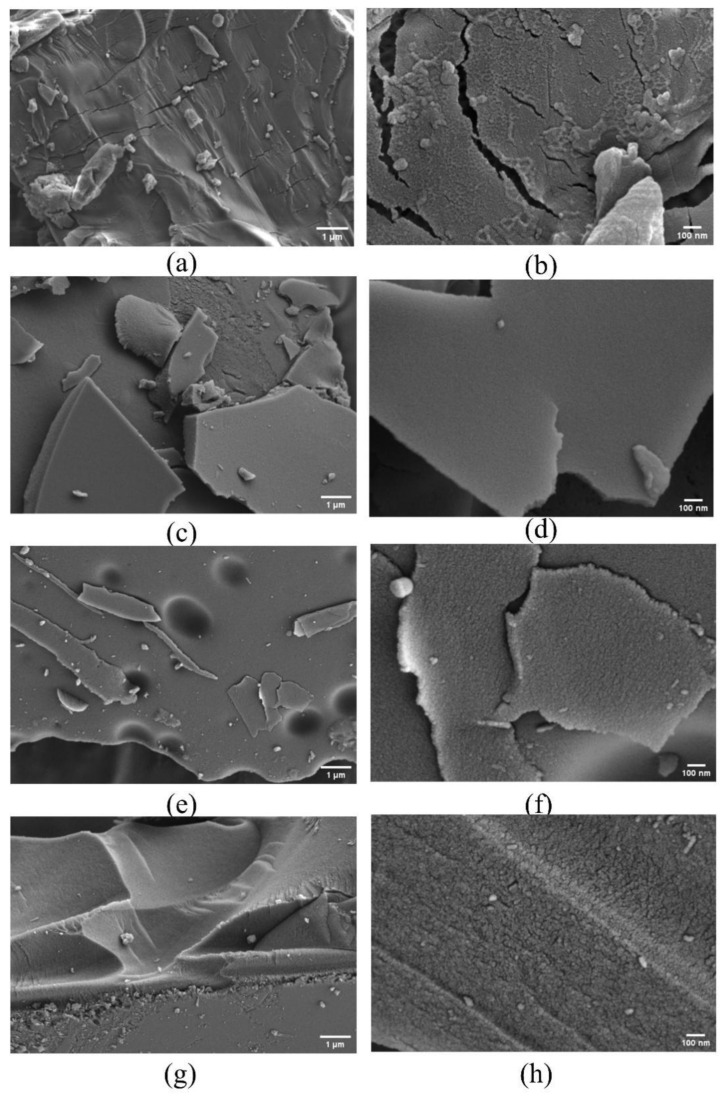
SEM images of β-CD samples (inactivated (**a**,**b**); activated at 400 °C (**c**,**d**); activated at 600 °C (**e**,**f**); activated at 800 °C (**g**,**h**)).

**Figure 7 molecules-27-07375-f007:**
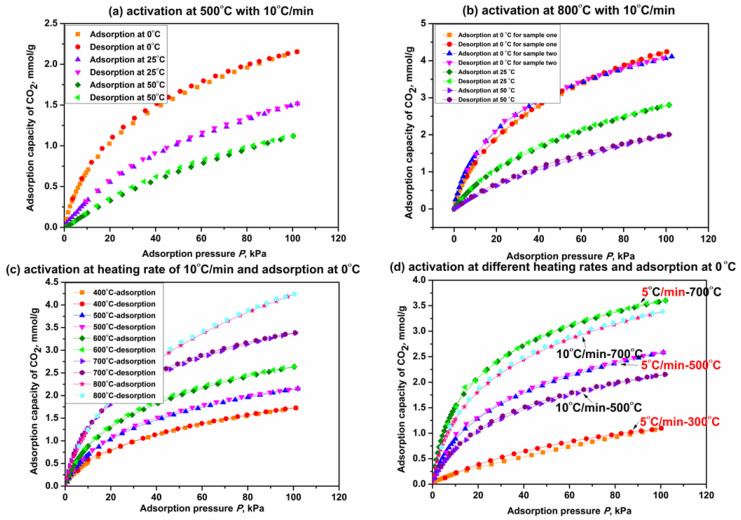
CO_2_ adsorption on β-CD derivatives at different adsorption temperatures (**a**,**b**); final activation temperatures (**c**) and heating rates (**d**).

**Table 1 molecules-27-07375-t001:** Function analysis of solid-state thermal decomposition model.

$g(\alpha )\;{}^{1}$	Rate Constant, k	Correlation Coefficient γ	Standard Deviation, δ	$g(\alpha )\;{}^{1}$	Rate Constant, k	Correlation Coefficient γ	Standard Deviation, δ
$D_{1}$	1.87 × 10^−4^	0.962	1.04 × 10^−6^	$R_{2}$	2.07 × 10^−4^	0.914	1.77 × 10^−6^
$D_{2}$	1.13 × 10^−4^	0.968	5.71 × 10^−6^	$R_{3}$	1.46 × 10^−4^	0.920	1.20 × 10^−6^
$D_{3}$	3.04 × 10^−5^	0.973	3.05 × 10^−5^	$A_{1}$	4.89 × 10^−4^	0.930	3.74 × 10^−6^
$D_{4}$	2.67 × 10^−5^	0.970	2.67 × 10^−5^	$A_{2}$	4.74 × 10^−4^	0.842	5.71 × 10^−6^
$A_{u}$	0.002	0.648	5.04 × 10^−5^	$A_{3}$	4.08 × 10^−4^	0.787	0.43
$R_{1}$	3.51 × 10^−4^	0.896	3.34 × 10^−6^	$A_{4}$	3.51 × 10^−4^	0.750	5.65 × 10^−6^
g(α)	**Correlation coefficient**		***E* (kJ/mol) ^2^**	g(α)	***E* (kJ/mol) ^3^**		
$D_{1}$	0.9873		99.1	$D_{1}$	102.7		
$D_{2}$	0.9904		104.7	$D_{2}$	106.4		
$D_{3}$	0.993		110.6	$D_{3}$	110.4		
$D_{4}$	0.9912		106.7	$D_{4}$	107.8		

^1^ the data of thermal activation at final activation temperature of 290 °C in Figure 1(b-2); ^2^ the data of thermal activation at final activation temperature of 290 °C, 310 °C and 330 °C in Figure 1(b-2–b-4);. ^3^ the data of thermal activation at final activation temperature of 500 °C with the heating rates of 5 °C/min, 10 °C/min, 15 °C/min and 20 °C/min, respectively.

**Table 2 molecules-27-07375-t002:** MS peaks and possible ion group fragments from gas products.

MS Peak, *m*/*z*	Intensity, *log*_10_(*A*)		Possible Ion Group Fragment
	5 °C/min	15 °C/min	
14	−10.18	−9.842	–^12^CH_2_–
16	−9.772	−9.871	–^16^O–
17	−8.754	−9.043	–^16^OH
18	−8.085	−8.355	H_2_^16^O(main), –^18^O–, –^16^OD
20 (main, leading peak)	−6.876	−6.91	H_2_D^16^O^+^ and H_2_^18^O, accompanied by H_3_^16^O^+^, –^18^OD, HD^16^O
28 with sub-peaks of 29 and 30	−8.693	−8.324	^12^C^16^O(main),^12^C_2_H_4_, accompanied by ^13^C^16^O, ^12^C^13^CH_4_, ^12^C_2_H_3_D, ^12^CH^16^O and ^12^C^18^O, ^12^CH_2_^16^O
32	−9.143	−8.862	^16^O_2_(main), ^12^CH_2_^18^O
36 with sub-peak of 34	−8.307	−8.354	H^16^O^18^OH, accompanied by H_2_^16^O_2_, ^16^O^18^O
38	−9.016	−9.037	–^12^C_3_H_2_–
40 (main, leading peak)	−5.81	−5.85	^12^C_3_H_4_(main), –^12^C_2_^13^CH_3_, –^12^C_3_H_2_D, accompanied by –^12^C_3_H_3_, –^12^C_2_^13^CH_2_–, –^12^C_3_HD–
44	−9.925	−10.2	^12^C^16^O_2_(main), ^12^CH_3_^12^CH^16^O
80	−11.05	−11.2	^12^C_6_H_8_, ^12^C_5_^16^OH_4_

**Table 3 molecules-27-07375-t003:** MS Fragments of original β-CD sample.

Item	Average Relative Mass	MS Observation	Difference	Possible Group Fragments
One glycosidic unit	162	183	21	e.g., one glycosidic unit containing one ^13^C atom, accompanied by one ^16^O atom at the location of C4 and the transfer of four protons
Two glycosidic units	324	447	M_162_–M_39_	M_39_(*C_3_H_3_, cyclopropene radical)
Three glycosidic units	486	559/600	73/113	M_73_(e.g., –CH(OH)–CH(OH)–*CH– or –*CH–O–*CH–CH2–OH), M_113_(–CH(O)–CH(O)–CH–CH=CH–OH)
Four glycosidic units	648	713	M_826_–M_113_	M_40_(C_3_H_4_, cyclopropene)
Five glycosidic units	810	826	16	M_16_–O
Six glycosidic units	972			
Seven glycosidic units	1134	1169	35	isotope and –OH

**Table 4 molecules-27-07375-t004:** Pore structural characteristics of β-CD sample before and after activation.

Sample	Specific Surface Area (m^2^/g)	Total Pore Volume (cm^3^/g)	Pore Volume of Micropores (cm^3^/g)	Average Pore Size (nm)	Most Probable Pore Size (nm)
Original β-CD	27.47	0.04	0.01	3.39	0.44
β-CD derivative with the final activation temperature of 500 °C	164.90	0.10	0.07	2.46	0.61
β-CD derivative with the final activation temperature of 800 °C	830.00	0.37	0.33	1.78	0.63

## Data Availability

Not applicable.

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
