# Peer review of "Porous Structure of β-Cyclodextrin for CO2 Capture: Structural Remodeling by Thermal Activation"

_molecules, 2022, doi:10.3390/molecules27217375_

Round 1

Reviewer 1 Report

Greetings, Editor thank you for providing me with the opportunity to review the article. I reviewed the article with ID =molecules-1985391. Overall, the article structure and content are suitable for the molecules journal. I am pleased to send you major level comments, there are some serious flaws which need to be corrected before publication. Please consider these suggestions as listed below.

  1. The title seems ok.   
  2. The abstract seems to be good. Please add one more introductory line of your objective in beginning of abstract.
  3. Research gap should be delivered on more clear way with directed necessity for the future research work.
  4. Introduction section must be written on more quality way, i.e., more up-to-date references addressed. Please target the specific gap such as 2015-2021 etc.
  5. The novelty of the work must be clearly addressed and discussed, compare previous research with existing research findings and highlight novelty.
  6. What is the main challenge?
  7. The length of introduction is very short. Please expand your discussion.
  8. In introduction Page 1 Line 31 need a reference please cite this article here- Yaqoob, A.A.; Ibrahim, M.N.M.; Ahmad, A.; Reddy, A.V.B. Toxicology and Environmental Application of Carbon Nanocomposite. In Environmental Remediation through Carbon Based Nano Composites; Springer: Berlin/Heidelberg, Germany, 2021; pp. 1–18.
  9. In introduction Page 1 Line 44 need another reference with existing 22. please cite this article here- Umar, K.; Yaqoob, A.A.; Ibrahim, M.N.M.; Parveen, T.; Safian, M.T. Environmental applications of smart polymer composites. Smart Polym. Nanocompos. Biomed. Environ. Appl. 202015, 295–320.
  10. The main objective of the work must be written on the more clear and more concise way at the end of introduction section.
  11. Please check the abbreviations of words throughout the article. All should be consistent.
  12. Please include all chemical/instrumentation brand name and other important specification.
  13. Please provide space between number and units. Please revise your paper accordingly since some issue occurs on several spots in the paper.
  14. Please use correct units such as temperature etc.
  15. Overall result section is well explained.
  16. Regarding the replications, authors confirmed that replications of experiment were carried out. However, these results are not shown in the manuscript, how many replicated were carried out by experiment? Results seem to be related to a unique experiment. Please, clarify whether the results of this document are from a single experiment or from an average resulting from replications. If replicated were carried out, the use of average data is required as well as the standard deviation in the results and figures shown throughout the manuscript. In case of showing only one replicate explain why only one is shown and include the standard deviations.
  17. The resolution of Figure 1,2,3,4 and 7 is very poor. Please split the images and deliver a clear. Its difficult to read it.
  18. Please add a comparative discussion section.
  19. Section 4 should be renamed by Conclusion and Future perspectives. Conclusion section is missing some perspective related to the future research work, quantify main research findings, highlight relevance of the work with respect to the field aspect.
  20. To avoid grammar and linguistic mistakes, major level English language should be thoroughly checked. Please revise your paper accordingly since several language issue occurs on several spots in the paper.
  21. Reference formatting need carefully revision. All must be consistent in one formate. Please follow the journal guidelines.

Author Response

Reviewer #1:  Greetings, Editor thank you for providing me with the opportunity to review the article. I reviewed the article with ID =molecules-1985391. Overall, the article structure and content are suitable for the molecules journal. I am pleased to send you major level comments, there are some serious flaws which need to be corrected before publication. Please consider these suggestions as listed below.

Response: Thanks for the reviewer’s comment. We will revise the paper carefully according to the following suggestions  and comments.

1.The title seems ok. 

Response: Thanks for the reviewer’s comment.The title has also been checked.

2.The abstract seems to be good. Please add one more introductory line of your objective in beginning of abstract.

Response: Thanks for the reviewer’s suggestion.The introductory line of my objective in beginning of abstract has been modified.

3.Research gap should be delivered on more clear way with directed necessity for the future research work.

Response: Thanks for the reviewer’s suggestion.The end of the abstract and introduction have been modified in order to clearly deliver the directed necessity for the future research work.

4.Introduction section must be written on more quality way, i.e., more up-to-date references addressed. Please target the specific gap such as 2015-2021 etc.  

Response: Thanks for the reviewer’s suggestion. According to the reviewer’s suggestion, about 30 up-to-date references were added, specially for the gap such as the year 2015-2021.

5.The novelty of the work must be clearly addressed and discussed, compare previous research with existing research findings and highlight novelty.

Response: Thanks for the reviewer’s suggestion.The novelty of the work has been clearly addressed and discussed by comparing previous research with existing research findings based on new more than 30 references.

6.What is the main challenge?

Response: Thanks for the reviewer’s comment.The main challenge is that the cavity structure of β-CD has been partially destroyed during the thermal activation. We will seek for the low-temperature activation method to avoid the cracking of the cavity structure of β-CD in our future studies.

7.The length of introduction is very short. Please expand your discussion.

Response: Thanks for the reviewer’s comment. According to the reviewer’s comment, the introduction has been expanded.

8.In introduction Page 1 Line 31 need a reference please cite this article here- Yaqoob, A.A.; Ibrahim, M.N.M.; Ahmad, A.; Reddy, A.V.B. Toxicology and Environmental Application of Carbon Nanocomposite. In Environmental Remediation through Carbon Based Nano Composites; Springer: Berlin/Heidelberg, Germany, 2021; pp. 1–18.

Response: Thanks for the reviewer’s suggestion.The article has been cited in the introduction.

9.In introduction Page 1 Line 44 need another reference with existing 22. please cite this article here- Umar, K.; Yaqoob, A.A.; Ibrahim, M.N.M.; Parveen, T.; Safian, M.T. Environmental applications of smart polymer composites. Smart Polym. Nanocompos. Biomed. Environ. Appl. 2020, 15, 295–320.

Response: Thanks for the reviewer’s suggestion.The reference has been cited in the introduction, but the format is modified as “Umar, K.; Yaqoob, A.A.; Ibrahim, M.N.M.; Parveen, T.; Safian, M.T. Environmental applications of smart polymer composites. In Smart Polymer Nanocomposites; Woodhead Publishing, 2021; Volume 13, pp. 295-312.”

10.The main objective of the work must be written on the more clear and more concise way at the end of introduction section.

Response: Thanks for the reviewer’s suggestion. The main objective of the work has been rewritten on the more clear and more concise way at the end of introduction section.

11.图片里Please check the abbreviations of words throughout the article. All should be consistent.

Response: Thanks for the reviewer’s suggestion.The abbreviations in the figures such as Figure 2 have been modified to be  consistent, and the abbreviations throughout the article have also been consistently checked.

12.Please include all chemical/instrumentation brand name and other important specification.

Response: Thanks for the reviewer’s suggestion.All the chemical/instrumentation brand name and other important specification have been added, specially in the Section 3.

13.Please provide space between number and units. Please revise your paper accordingly since some issue occurs on several spots in the paper.

Response: Thanks for the reviewer’s suggestion.The space between number and unit has added.

14.Please use correct units such as temperature etc.

Response: Thanks for the reviewer’s comment.The units used in the paper have been modified. For example, the temperature unit is consistently modified as degrees celsius (℃).

15.Overall result section is well explained.

Response: Thanks for the reviewer’s comment. Your comment will encourage us to get more achievements in the future.

16.Regarding the replications, authors confirmed that replications of experiment were carried out. However, these results are not shown in the manuscript, how many replicated were carried out by experiment? Results seem to be related to a unique experiment. Please, clarify whether the results of this document are from a single experiment or from an average resulting from replications. If replicated were carried out, the use of average data is required as well as the standard deviation in the results and figures shown throughout the manuscript. In case of showing only one replicate explain why only one is shown and include the standard deviations.

Response: Thanks for the reviewer’s comment. The comment is very important for us to obtain reliable data in our future researches. According to the comment, the isotherm of CO2 adsorption on the  β-CD deriviate parallelly activated at 800℃ has been added in Figure 7. And the capacity error at 0℃ and below 100kPa is below 0.2 mmol/g based on two parallell samples. Actually, in this paper, those conclusions were obtained based on the qualitative analysis of comparison data. The result of FTIR of every test is obtained based on 16 scans,and the error is below 0.1%. The thermochemical charateristics at the third stage were verified based on the four tests with the final activation temperatures of 270, 290, 310, 330 ℃. And the variation tendencies of surface contents such as carbon and oxygen were verified based on the test results at three β-CD deriviates activated at 400, 600, 800℃, respectively. Then the effect of heating rate on capacity was verified by comparing the isotherm at 10 ℃/min with the isotherm at 5 ℃/min based on two different activation temperatures of 700 ℃ and 500 ℃, not the replicates of the isotherm at 10 ℃/min with the isotherm at 5 ℃/min based on a given activation temperature of 700 ℃ or 500 ℃. And so on.

17.The resolution of Figure 1,2,3,4 and 7 is very poor. Please split the images and deliver a clear. Its difficult to read it.

Response: Thanks for the reviewer’s suggestion.The Figures 1,2,3,4 and 7 have been modified by different colors and  distinguishable lines, accompanied by the resolution of  600 DPI.

18.Please add a comparative discussion section.

Response: Thanks for the reviewer’s comment.The related comparative discussion section has been added based on new more than 30 references.

19.Section 4 should be renamed by Conclusion and Future perspectives. Conclusion section is missing some perspective related to the future research work, quantify main research findings, highlight relevance of the work with respect to the field aspect.

Response: Thanks for the reviewer’s suggestion. The section 4 has been renamed as “Conclusion and Future perspectives”, And some perspective related to the future research work, quantifying main research findings, highlighting the  relevance of the work with respect to the field aspect have been added.

20.To avoid grammar and linguistic mistakes, major level English language should be . Please revise your paper accordingly since several language issue occurs on several spots in the paper.

Response: Thanks for the reviewer’s suggestion. According to the suggestion, the paper has been accordingly revised to avoid the grammar and linguistic mistakes.

21.Reference formatting need carefully revision. All must be consistent in one formate. Please follow the journal guidelines.

Response: Thanks for the reviewer’s suggestion.The reference formatting has been carefully revised and all are consistent in one format.

Reviewer 2 Report

The authors studied mechanism of a  adsorbent of β-cyclodextrin aggregates for CO2 capture. They used  TG-DSC, in situ TG-FTIR/FTIR, TG-MS/MS, EDS, XPS DFT experiments and calulations to obtain the  information obout the thermochemical characteristics during pore formation of β-cyclodextrin organic frameforks. This subject is relevant to gain a novel understanding of the fitures of porous structure of these system. I believe the authors made a nice sound work. However, the introduction and discussion do not include citations from relevant articles by key scientists in the field of materials of β-cyclodextrin and CO2 capture. It deprivation may create an impression about the low awareness of the authors. In this regard, please cite the works from the list of publications doi below:
- application β-cyclodextrin in drug delivery for ondroduction
10.1016/j.molliq.2022.119548
10.1007/s10973-017-6252-1
and
-comparative analysis CO2 capture data of other materials MOFs and model polymers
 10.1021/ja0570032
 10.1134/S1990793121070101

Minor issues:
Make the caption font size larger in the Figure 1,2,3 and 7
Add a scale bar to Figure 6 for SEM picture.

Author Response

Dear the reviewer,

Manuscript ID molecules-1985391 entitled "Porous structure of β-cyclodextrin for CO2 capture: structural remodeling by thermal activation" submitted to Molecules has been revised. The detailed/point-by-point responses to the reviewers' comments are listed in the following.

 MDPI/ Molecules  

PAPER:  molecules-1985391_R1 

TITLE: Porous structure of β-cyclodextrin for CO2 capture: structural remodeling by thermal activation 

 Report of Review

Reviewer comments:    

Reviewer #2:  

1.The authors studied mechanism of a  adsorbent of β-cyclodextrin aggregates for CO2 capture. They used  TG-DSC, in situ TG-FTIR/FTIR, TG-MS/MS, EDS, XPS DFT experiments and calulations to obtain the  information obout the thermochemical characteristics during pore formation of β-cyclodextrin organic frameforks. This subject is relevant to gain a novel understanding of the fitures of porous structure of these system. I believe the authors made a nice sound work. However, the introduction and discussion do not include citations from relevant articles by key scientists in the field of materials of β-cyclodextrin and CO2 capture. It deprivation may create an impression about the low awareness of the authors. In this regard, please cite the works from the list of publications doi below:

- application β-cyclodextrin in drug delivery for ondroduction

10.1016/j.molliq.2022.119548,10.1007/s10973-017-6252-1 and

-comparative analysis CO2 capture data of other materials MOFs and model polymers

 10.1021/ja0570032, 10.1134/S1990793121070101

Response: Thanks for the reviewer’s suggestion.The relevant articles have been cited in the paper.

Reviewer #3: Minor issues:

2.Make the caption font size larger in the Figure 1,2,3 and 7

Response: Thanks for the reviewer’s suggestion.The caption font size in the Figure 1,2,3 and 7 has been enlarged.

3.Add a scale bar to Figure 6 for SEM picture.

Response: Thanks for the reviewer’s suggestion.The scale bar has been added to Figure 6 for SEM picture.

Round 2

Reviewer 1 Report

Accepted in the present form